# Multilocus Phylogeny and Characterization of Five Undescribed Aquatic Carnivorous Fungi (*Orbiliomycetes*)

**DOI:** 10.3390/jof10010081

**Published:** 2024-01-20

**Authors:** Fa Zhang, Yao-Quan Yang, Fa-Ping Zhou, Wen Xiao, Saranyaphat Boonmee, Xiao-Yan Yang

**Affiliations:** 1Institute of Eastern-Himalaya Biodiversity Research, Dali University, Dali 671003, China; zhangf@eastern-himalaya.cn (F.Z.); yanyq@eastern-himalaya.cn (Y.-Q.Y.); zhoufp@eastern-himalaya.cn (F.-P.Z.); xiaow@eastern-himalaya.cn (W.X.); 2Center of Excellence in Fungal Research, Mae Fah Luang University, Chiang Rai 57100, Thailand; 3School of Science, Mae Fah Luang University, Chiang Rai 57100, Thailand; 4The Provincial Innovation Team of Biodiversity Conservation and Utility of the Three Parallel Rivers Region, Dali University, Dali 671003, China; 5Yunling Back-and-White Snub-Nosed Monkey Observation and Research Station of Yunnan Province, Dali 671003, China

**Keywords:** aquatic habitat, *Arthrobotrys*, carnivorous fungi, new species, *Orbiliaceae*, phylogeny

## Abstract

The diversity of nematode-trapping fungi (NTF) holds significant theoretical and practical implications in the study of adaptive evolution and the bio-control of harmful nematodes. However, compared to terrestrial ecosystems, research on aquatic NTF is still in its early stages. During a survey of NTF in six watersheds in Yunnan Province, China, we isolated 10 taxa from freshwater sediment. Subsequent identification based on morphological and multigene (*ITS*, *TEF1-α*, and *RPB2*) phylogenetic analyses inferred they belong to five new species within *Arthrobotrys*. This paper provides a detailed description of these five novel species (*Arthrobotrys cibiensis*, *A. heihuiensis*, *A. jinshaensis*, *A. yangbiensis*, and *A. yangjiangensis*), contributing novel insights for further research into the diversity of NTF and providing new material for the biological control of aquatic harmful nematodes. Additionally, future research directions concerning aquatic NTF are also discussed.

## 1. Introduction

Nematode-trapping fungi (NTF) are a group of fungi that possess a unique trapping structure to capture nematodes for nutrition [1,2,3,4]. NTF in *Orbiliomycetes* are considered the core representatives of NTF due to their rich species diversity, and intricate and diverse trapping structures, as well as their important role in maintaining ecological balance and their potential value in the bio-control of harmful nematodes [4,5,6,7]. Currently, this group of fungi includes 125 species from three genera: *Arthrobotrys* (73 species) which captures nematodes using adhesive networks; *Dactylellina* (35 species), the genus that captures nematodes with adhesive branches, non-constricting rings, and adhesive knobs; and *Drechslerella* (17 species) which catches nematodes using constricting rings [4,7,8].

These fungi are widely distributed in various habitats because of their unique survival strategy. They are commonly found in the soils from farmlands, forests, and even heavy metal-contaminated areas [4,9,10,11], as well as in sediments from marine, freshwater, and even hot springs [12,13,14]. But compared to the well-studied terrestrial ecosystems, the diversity of NTF in freshwater habitats remains insufficiently studied [13,15,16]. Previous studies have confirmed the existence of a rich diversity of NTF in freshwater ecosystems, which is reasonable given the abundance of nematodes in aquatic environments [13,17]. Meanwhile, the diverse array of nematodes in aquatic habitats includes parasitic species that pose threats to aquatic crops and fisheries [18,19]. So, studying aquatic NTF resources is an important part of NTF diversity research and bio-control of harmful aquatic nematodes. Additionally, the study on aquatic NTF also provides a valuable entry point for investigating fungal adaptive evolution, as aquatic NTF originate from their terrestrial counterparts.

In the past 10 years, we have investigated the NTF in the six major watersheds in Yunnan Province and successfully isolated 10 strains, which were identified as five novel members of *Arthrobotrys*. This paper provides a comprehensive account of these species, offering new material for the bio-control research of harmful nematodes and the study of fungal aquatic adaptive evolution.

## 2. Materials and Methods

### 2.1. Samples Collection

All freshwater sediment samples involved in this study were collected using a Peterson bottom sampler (HL-CN, Wuhan Hengling Technology Company, Limited, Wuhan, China). The samples were placed into plastic zip-lock bags to preserve moisture. Collecting sites, date, and collector were recorded (Table 1). The samples were stored at 4 °C and processed within a week.

### 2.2. Fungal Isolation

Nematodes (*Panagrellus redivivus* Goodey, free-living nematodes) cultured on oatmeal medium [4] were isolated using the Baermann funnel method [20] and the concentration of the nematodes was adjusted with sterile water to 3000–5000 nematodes per milliliter. The soil sprinkling technique was used to disperse the sediment sample onto the surface of corn meal agar plates (CMA) [4] and 1 mL of nematode suspension was added to promote the germination of NTF. The plates were incubated at room temperature (14–28 °C) for about three weeks and a stereomicroscope was used to observed the plates to search for the NTF spores. The single-spore isolation method was used for the isolation and purification of the NTF [4].

### 2.3. Morphological Observation

The isolates were inoculated onto potato dextrose agar (PDA) [4] plates and cultured at 26 °C for colony observation. The isolate was transferred to CMA observation plates (creating an observation well by removing a 2 × 2 cm piece of agar from the center of the CMA plate and obliquely inserting a sterile cover glass into the surface of the medium) and incubated at 26 °C [21]. After the observation well was covered by the mycelium, about 1000 nematodes (*P. redivivus*) were added as baits to induce the production of traps. The types of traps were checked and photographed using an Olympus BX53 microscope (Olympus Corporation, Tokyo, Japan). When the mycelium had spread over the cover glass, the cover glass was removed with tweezers and a temporary slide made with sterile water [22]. An Olympus BX53 microscope (Olympus Corporation, Tokyo, Japan) was used to photograph and measure the morphological characteristics such as conidia, conidiophores, and chlamydospores.

### 2.4. Collection of DNA Molecular Data

The mycelium grown on PDA plates was used to extract genomic DNA, as described by Zhang et al. [15]. The primer pairs ITS4-ITS5 [23], 526F-1567R [24], and 6F-7R [25] were used to amplify the *ITS*, *TEF1-α*, and *RPB2* regions, respectively, under the reaction system and conditions described in the previous study [21]. The PCR products were sent to BioSune Biotech Company Limited (Shanghai, China) for purification and sequencing (*TEF1-α* genes were sequenced using the 247F-609R [7] primer pair and *ITS* and *RPB2* regions were sequenced with PCR primers).

The generated sequences were carefully examined, edited, and assembled using SeqMan v. 7.0 [26]. All sequences obtained in this study have been submitted to the GenBank database (NCBI; https://www.ncbi.nlm.nih.gov/; accessed on 29 November 2023) for deposition.

### 2.5. Phylogenetic Analysis

The sequences generated in this study were compared with the GenBank database using BLASTn (https://blast.ncbi.nlm.nih.gov/; accessed on 9 November 2023). Our five species were placed within *Arthrobotrys* according to the BLASTn search and their trapping structures [4]. Consequently, relevant publications [3,4,7,15,21,27] and the BLASTn search results were used to retrieve all reliable *ITS*, *TEF1-α*, and *RPB2* sequences of *Arthrobotrys* taxa from the GenBank database (Appendix A). Three genes were aligned via the online program MAFFT v. 7 (http://mafft.cbrc.jp/alignment/server/; accessed on 14 November 2023) [28]. MEGA6.0 [29] was used to adjust and link the three alignments.

The best-fit optimal substitution models for *ITS* (GTR + I + G), *TEF1-α* (SYM + I + G), and *RPB2* (SYM + I + G) were calculated using jModelTest v. 2.1.10 [30].

Two *Vermispora* species (*V. fusarina* (YXJ02-13-5) and *V. leguminacea* (CGMCC 6.0291)) were set as outgroups. IQ-Tree v. 1.6.5 [31] and MrBayes v. 3.2.6 [32] were used to infer the phylogenetic trees using maximum likelihood (ML) and Bayesian inference (BI) methods. The related parameter settings are the same as in the previous study [21].

The trees were visualized via FigTree v. 1.3.1 [33] and edited using Microsoft PowerPoint v. 2016 (Microsoft, Redmond, WA, USA) and Adobe Photoshop CS6 software (Adobe Systems, San Jose, CA, USA).

## 3. Results

### 3.1. Phylogenetic Analysis

The combined *ITS*, *TEF1-α*, and *RPB2* alignment dataset consisted of 88 sequences of *ITS*, 62 sequences of *TEF1-α*, and 64 sequences of *RPB2* from *Arthrobotrys* 75 taxa, representing 69 valid species (plus our five new species), other related taxa in *Orbiliomycetes* (*Dactylellina* four taxa and *Drechslerella* seven taxa), and two outgroup taxa. The final dataset comprised 2000 characters (585 for *ITS*, 8321 for *RPB2*, and 583 for *TEF1-α*), among which 900 base pair (bp) are constant, 1087 bp are variable, and 886 bp are parsimony-informative.

The best-scoring ML tree was generated with a final ML optimization likelihood value of −6817.314758. Bayesian analysis (BI) was used to evaluate the Bayesian posterior probabilities with a final average standard deviation of the split frequency of 0.009092. Both ML and BI trees consistently grouped all tested nematode-trapping fungi into three major clades and five new species exhibited distinct divergence from known species. Therefore, the ML tree was chosen for presentation (Figure 1).

The phylogenetic tree inferred from the *ITS*, *TEF1-α*, and *RPB2* combined dataset placed five new species in *Arthrobotrys*. The phylogenetic position of *A. heihuiensis* is uncertain but clearly diverges from known species. The two isolates of *A. yangbiensis* formed a distinct lineage basal to *A. gampsospora* with 99% MLBS and 0.97 BYPP support. Furthermore, *A. yangjiangensis* and *A. jinshaensis* clustered together with *A. mangrovispora, A. thaumasia, A. eudermata*, and *A. janus* with 89% MLBS and 0.91 BYPP support. *A. cibiensis* formed a distinct lineage basal to *A. longiphora, A. xiangyunensis*, and *A. reticulatus* with 99% MLBS and 0.96 BYPP support (Figure 1).

### 3.2. Taxonomy

#### 3.2.1. *Arthrobotrys cibiensis* F. Zhang, S. Boonmee, and X.Y. Yang sp. nov. (Figure 2)

Index Fungorum number: IF901486; Facesoffungi number: FOF14174

Etymology: The species name “*cibiensis*” refers to the name of the sample collection site: Cibi Lake, Eryuan County, Dali City, Yunnan Province, China.

Material examined: China, Yunnan Province, Dali City, Eryuan County, Cibi Lake, 26°9′7.14″ N, 99°56′32.72″ E, from freshwater sediment, 4 June 2013, F. Zhang. Holotype CGMCC 3.20970, preserved in the China General Microbiological Culture Collection Center. Ex-type culture DLUCC 109, preserved in the Dali University Culture Collection. Additional specimen examined: China, Yunnan Province, Dali City, Eryuan County, Cibi Lake, 26°9′7.14″ N, 99°56′32.72″ E, from freshwater sediment, 4 June 2013, F. Zhang. Living culture EY10, preserved in germplasm resources center of Institute of Eastern-Himalaya Biodiversity Research, Dali University.

Culture characteristics: *Colonies* on PDA white, cottony, reaching 60 mm diameter after 10 days at 26 °C. *Hypha* composed of septate, branched, smooth, and hyaline. *Conidiophores* erect, septate, hyaline, unbranched, bearing a single conidium at the apex, 145–315.5 µm long ($\bar{X}$ = 234.4 µm, n = 50), 4.5–7.5 µm ($\bar{X}$ = 5.6 µm, n = 50) wide at the base, and 2–4 µm ($\bar{X}$ = 3.1 µm, n = 50) wide at the apex. *Conidia* smooth-faced and hyaline, rounded at the apex and truncated at the base, 26.5–46 × 13.5–23 µm ($\bar{X}$ = 37.1 × 17.7 μm, n = 50), immature having drop-shaped, obovate, with a super cell (the cell in the conidia significantly larger than other cells) at the apex and one to three septa (mostly two-septate) at the base of the conidia; mature conidia subfusiform, two- to three-septate (mostly three-septate, one septum at the apex and two septa at the base), with a super cell at the middle of the conidia. Conidia germinate from the small cells at both ends and the super cells never germinate. Catching nematodes with *adhesive networks*. *Chlamydospores* 9–35.5 × 6.5–13 µm ($\bar{X}$ = 16.2 × 10.1 μm, n = 50), smooth-faced and hyaline, cylindrical, globose or ellipsoidal, hyaline, and in chains when present.

Notes: The phylogenetic analyses revealed that *A. cibiensis* are grouped within a clade of *A. longiphora*, *A. reticulatus*, and *A. xiangyunensis* with 99% MLBS and 0.96 BYPP support (Figure 1). A comparison of the *ITS* nucleotides indicates that *A*. *cibiensis* differs from *A. longiphora* (11.8% (56/473 bp)), *A. reticulatus* (3.0% (25/834 bp)), and *A. xiangyunensis* (3.1% (24/768 bp)), respectively. However, *A*. *cibiensis* produces conidia with no more than three septa, while *A. longiphora*, *A. xiangyunensis*, and *A. reticulatus* produce four- or five-septate conidia. In addition, the conidia of *A. cibiensis* are significantly smaller than these three species (*A. cibiensis*, 26.5–46 (37.1) × 13.5–23 (17.7) µm versus *A. longiphora*, 40–90 (54) × 15–27.5 (18) µm versus *A. xiangyunensis*, 27–72 (55.8) × 14.5–28.5 (21.9) µm versus *A. reticulatus*, 50–65 × 20–25 µm) [4,12,34].

**Figure 2 jof-10-00081-f002:**
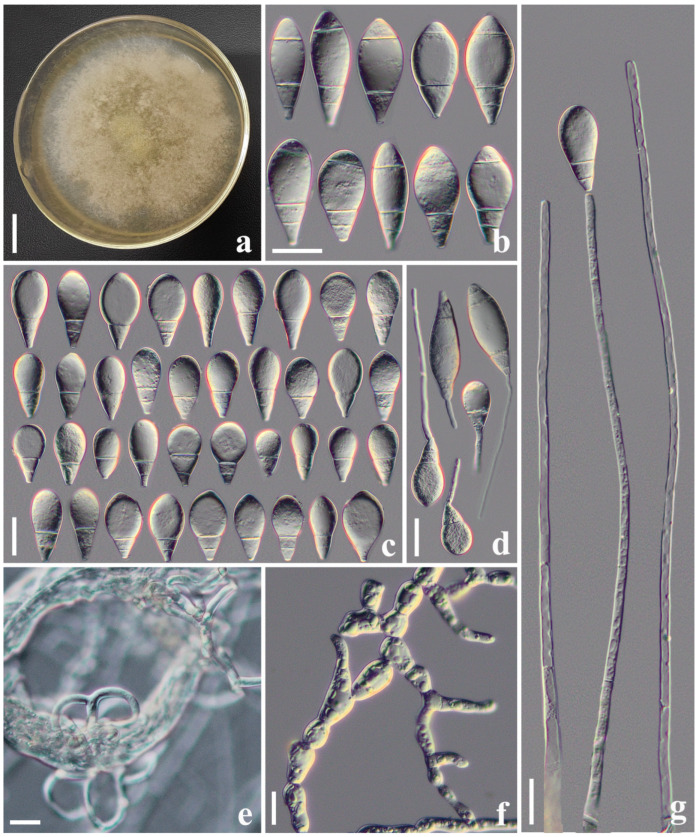
*Arthrobotrys cibiensis* (CGMCC 3.20970). (**a**) Colony. (**b**) Mature conidia. (**c**) Immature conidia. (**d**) Germinating conidia. (**e**) Adhesive networks. (**f**) Chlamydospores. (**g**) Conidiophores. Scale bars: (**a**) = 1 cm, (**b**–**g**) = 20 μm.

#### 3.2.2. *Arthrobotrys heihuiensis* F. Zhang, S. Boonmee, and X.Y. Yang sp. nov. (Figure 3)

Index Fungorum number: IF901485; Facesoffungi number: FOF14175

Etymology: The species name “*heihuiensis*” refers to the Heihui River, the alias of the Yangbi River, where the species was first collected.

Material examined: China, Yunnan Province, Dali City, Yangbi County, Heihui River, 25°37′4.13″ N, 100°1′52.06″ E, from freshwater sediment, 6 April 2018, F. Zhang. Holotype CGMCC 3.20967, preserved in the China General Microbiological Culture Collection Center. Ex-type culture DLUCC 108-1, preserved in the Dali University Culture Collection. Additional specimen examined: China, Yunnan Province, Dali City, Yangbi County, Heihui River, 25°37′4.13″ N, 100°1′52.06″ E, from freshwater sediment, 6 April 2018, F. Zhang. Living culture Y710, preserved in germplasm resources center of Institute of Eastern-Himalaya Biodiversity Research, Dali University.

Culture characteristics: *Colonies* on PDA white, cottony, reaching 55 mm diameter after 10 days at 26 °C. *Hypha* composed of septate, branched, smooth, hyaline. *Conidiophores* erect, septate, unbranched or occasionally producing a long branch, each branch produces several clusters of short denticles (1–3 denticles, mostly 1) by repeated elongation, each short denticle bear a single conidium, 360–720 µm long ($\bar{X}$ = 561.3 µm, n = 50), 3.5–6.5 µm ($\bar{X}$ = 5 µm, n = 50) wide at the base, 1.5–3.5 µm ($\bar{X}$ = 2.3 µm, n = 50) wide at the apex. *Conidia* two types: *macroconidia* 31–56 × 6.5–14.5 µm ($\bar{X}$ = 45.4 × 11.5 μm, n = 50), hyaline, clavate or subfusiform, rounded at the apex, constricted and truncated at the base, two- to five-septate (mostly three-septate, one septum at the apex and two septa at the base), with a super cell at the middle. *Microconidia* 18.5–26.5 × 5–9.5 µm ($\bar{X}$ = 20.2 × 6.7 μm, n = 50), hyaline, smooth-faced, clavate, drop-shaped or lagenate, rounded at the apex and truncated at the base, zero- to two-septate (mostly zero- or one-septate), producing with micro-cycle conidiation. Macroconidia germinate from the small cells at both ends, and the super cells never germinate. Catching nematodes with *adhesive networks*. *Chlamydospores* 7–30 × 6–12.5 µm ($\bar{X}$ = 16.2 × 9.3 μm, n = 50), hyaline, smooth-faced, cylindrical, globose or ellipsoidal, in chains when present.

Notes: Phylogenetically, *A. heihuiensis* is sister to *A. cystosporia* but lacking in statistical support (Figure 1). Of all *Arthrobotrys* species, the morphological characteristics of *A. heihuiensis* are relatively unique and only *A. scaphoides* may be confused with *A. heihuiensis* but there are several obvious differences between them: (1) the conidia of *A. heihuiensis* usually scattered on the short denticles of the conidiophores, while the conidia of *A. scaphoides* are usually clustered on the node of the short branch produced by conidiophores; and (2) *A. scaphoides* produces five- or six-septate conidia, while the conidia of *A. heihuiensis* have no more than four septa. Additionally, the conidia of *A. heihuiensis* are obviously smaller than that of *A. scaphoides* (*A. heihuiensis*, 31–56 (45.4) × 6.5–14.5 (11.5) µm versus *A. scaphoides*, 36.6–79.3 (57) × 11–17.5 (14) µm); (3) *A. heihuiensis* produces clavate, drop-shaped or lagenate, zero- to two-septate microconidia, while *A. scaphoides* does not produce microconidia [4,35,36].

**Figure 3 jof-10-00081-f003:**
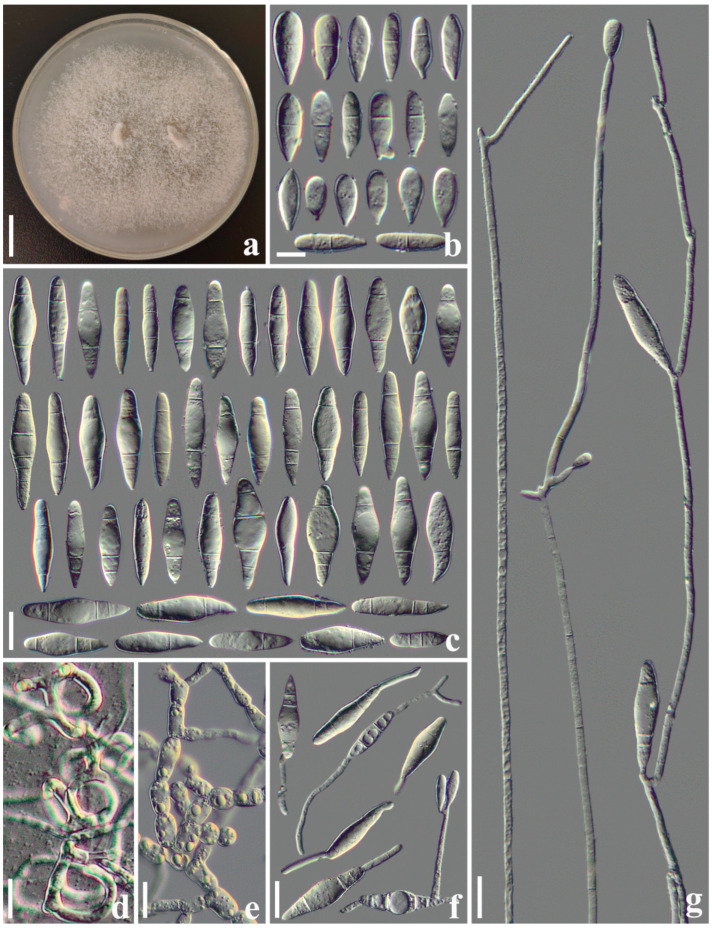
*Arthrobotrys heihuiensis* (CGMCC 3.20967). (**a**) Colony. (**b**) Microconidia. (**c**) Macroconidia. (**d**) Adhesive networks. (**e**) Chlamydospores. (**f**) Germinating conidia. (**g**) Conidiophores. Scale bars: (**a**) = 1 cm, (**b**) = 10 µm, (**c**–**g**) = 20 µm.

#### 3.2.3. *Arthrobotrys jinshaensis* F. Zhang, S. Boonmee and X.Y. Yang sp. nov. (Figure 4)

Index Fungorum number: IF901489; Facesoffungi number: FOF14176

Etymology: The species name “*jinshaensis*” refers to the name of sample collection site: Jinsha River, Jinjiang Town, Shangri-La City, Yunnan Province, China.

Material examined: China, Yunnan Province, Shangri-La City, Jinjiang Town, Jinsha River, 27°8′50.56″ N, 99°49′39.43″ E, from freshwater sediment, 9 July 2014, F. Zhang. Holotype CGMCC 3.20969, preserved in the China General Microbiological Culture Collection Center. Ex-type culture DLUCC 133, preserved in the Dali University Culture Collection. Additional specimen examined: China, Yunnan Province, Shangri-La City, Jinjiang Town, 27°8′50.56″ N, 99°49′39.43″ E, from freshwater sediment, 9 July 2014, F. Zhang. Living culture MA142, preserved in germplasm resources center of Institute of Eastern-Himalaya Biodiversity Research, Dali University.

Culture characteristics: *Colonies* on PDA white, cottony, reaching 55 mm diameter after 10 days at 26 °C. *Hypha* composed of septate, branched, smooth, hyaline. *Conidiophores* erect, septate, hyaline, unbranched or occasionally producing one to four short branches near the apex and bearing one to four conidia, 130–357 µm long ($\bar{X}$ = 246.3 µm, n = 50), 3.5–6 µm ($\bar{X}$ = 4.8 µm, n = 50) wide at the base, 2–4 µm ($\bar{X}$ = 3.1 µm, n = 50) wide at the apex. *Conidia* smooth-faced and hyaline, rounded at the apex and truncated at the base, 14.5–57 × 7.5–23 µm ($\bar{X}$ = 36.6 × 14.7 μm, n = 50), immature conidia drop-shaped, obovate, clavate, with a super cell at the apex and one to two septa (mostly two-septate) at the base; mature conidia subfusiform, clavate, two- to four-septate (mostly three-septate, one septum located at the apex and two septa at the base of the conidia), with a super cell at the middle of the conidia. Conidia germinate from the small cells at both ends, and the super cells never germinate. Catching nematodes with *adhesive networks*. *Chlamydospores* 9.5–21.5 × 5.5–9 µm ($\bar{X}$ = 14.8 × 7.2 μm, n = 50), smooth-faced and hyaline, cylindrical, globose or ellipsoidal, in chains when present.

Notes: Phylogenetically, *A. jinshaensis* clusters together with *A. yangjiangensis*, *A. mangrovispora*, *A. thaumasia*, *A. eudermata*, and *A. janus* with 89% MLBS and 0.91 BYPP support (Figure 1). There are 7.9% (49/622 bp), 7.3% (43/586 bp), 7.6% (46/607 bp), 8.2% (50/611 bp), and 8% (46/577 bp) differences between *A*. *jinshaensis* and *A. yangjiangensis*, *A. mangrovispora*, *A. thaumasia*, *A. eudermata*, and *A. janus* in *ITS* sequences, respectively. Morphologically, *A*. *jinshaensis*, *A. mangrovispora*, *A. megalospora*, *A. microscaphoides*, *A. obovata*, *A. oudemansii*, and *A. psychrophila* all produced conidiophores with short branches at the apex. However, *A*. *jinshaensis* can be easily distinguished from *A. megalospora*, *A. microscaphoides*, *A. obovata*, *A. oudemansii*, and *A. psychrophila* by its clavate, subfusiform, irregularly constricted, two- to four-septate conidia [4]. In comparison, *A. jinshaensis* is more similar to *A. mangrovispora* in its variable conidia. The main difference between *A. jinshaensis* and *A. mangrovispora* is that the conidia of *A. jinshaensis* are one- to four-septate and mostly three-septate, while the conidia of *A. mangrovispora* are one- to three-septate and mostly two-septate. Furthermore, a single conidiophores of *A. mangrovispora* may bear one to six conidia, while the conidiophore of *A*. *jinshaensis* bear no more than four conidia [4,37].

**Figure 4 jof-10-00081-f004:**
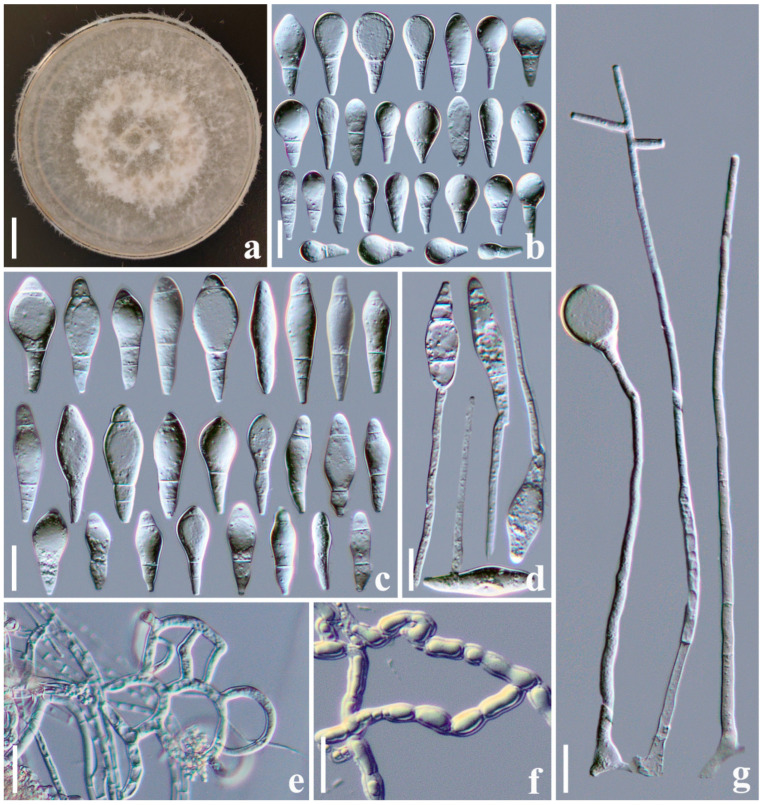
*Arthrobotrys jinshaensis* (CGMCC 3.20969). (**a**) Colony. (**b**) Immature conidia. (**c**) Mature conidia. (**d**) Germinating conidia. (**e**) Adhesive networks. (**f**) Chlamydospores. (**g**) Conidiophores. Scale bars: (**a**) = 1 cm, (**b**–**g**) = 20 µm.

#### 3.2.4. *Arthrobotrys yangbiensis* F. Zhang, S. Boonmee, and X.Y. Yang sp. nov. (Figure 5)

Index Fungorum number: IF901487; Facesoffungi number: FOF14177

Etymology: The species name “*yangbiensis*” refers to the name of sample collection site: Yangbi County, Dali City, Yunnan Province, China.

Material examined: China, Yunnan Province, Dali City, Yangbi County, Yangbi River, 25°42′37.94″ N, 99°54′52.15″ E, from freshwater sediment, 4 April 2018, F. Zhang. Holotype CGMCC 3.24985, preserved in the China General Microbiological Culture Collection Center. Ex-type culture DLUCC 36-1, preserved in the Dali University Culture Collection. Additional specimen examined: China, Yunnan Province, Dali City, Yangbi County, Yangbi River, 25°42′37.94″ N, 99°54′52.15″ E, from freshwater sediment, 4 April 2018, F. Zhang. Living culture Y678, preserved in germplasm resources center of Institute of Eastern-Himalaya Biodiversity Research, Dali University.

Culture characteristics: *Colonies* on PDA white, cottony, reaching 40 mm diameter after 10 days at 26 °C. *Hypha* composed of septate, branched, smooth, and hyaline. *Conidiophores* erect, septate, hyaline, unbranched or sometimes branched at the upper half part, producing a cluster of short denticles (2–5) at the apex or producing several clusters of short denticles by repeated elongation, 210–365 µm long ($\bar{X}$ = 284.7 µm, n = 50), 3–5.5 µm ($\bar{X}$ = 3.9 µm, n = 50) wide at the base, and 1.5–3.5 µm ($\bar{X}$ = 2.4 µm, n = 50) wide at the apex. *Conidia* 40.5–73 × 8.5–18 µm ($\bar{X}$ = 55.4 × 13.6 μm, n = 50), elongate–fusiform or clavate, some conidia curved, hyaline, smooth-faced, two- to five-septate (mostly three- or four-septate), with a super cell at the middle or apex of the conidia, and some conidia produce coiled filamentous appendages. Conidia germinate from the small cells at both ends and the super cells never germinate. They capture nematodes with *adhesive networks*. *Chlamydospore* not observed.

Notes: The phylogenetic analyses revealed that *A. yangbiensis* is sister to *A. gampsospora* with 99% MLBS and 0.97 BYPP support (Figure 1). *A. yangbiensis* is 14.7% (118/798 bp) different from *A. gampsospora* in *ITS* sequences. In morphology, *A. yangbiensis* shares some morphological features of short denticle conidiophores and sub-fusiform, curved conidia with *A. gampsospora* but they differ in shape and size of microconidia. In addition, the conidiophores of *A. yangbiensis* produce several clusters of short denticles by repeated elongation or geniculate branches of conidiophores, and bear several clusters of conidia, while *A. gampsospora* bears a single cluster of conidia at the apex of conidiophores [4,38].

**Figure 5 jof-10-00081-f005:**
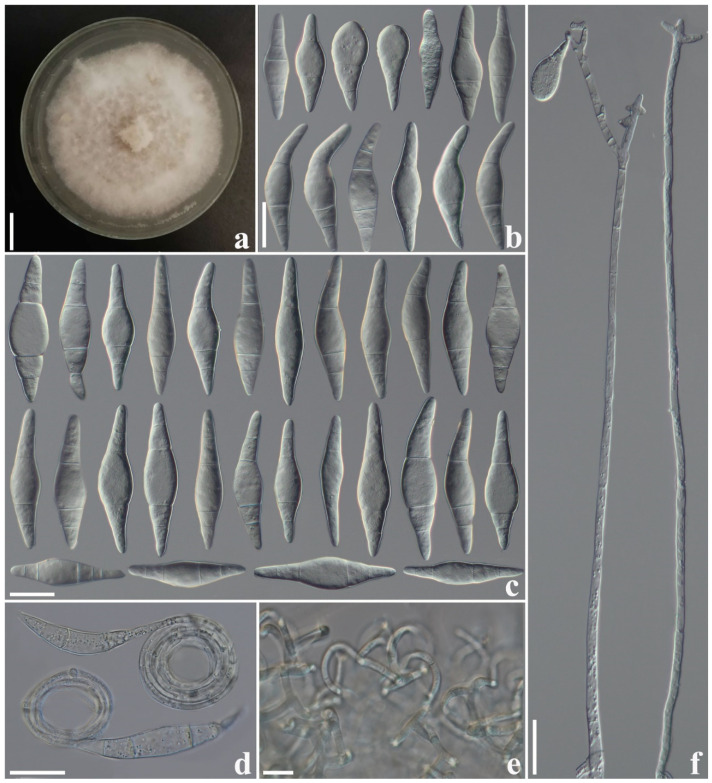
*Arthrobotrys yangbiensis* (CGMCC 3.20943). (**a**) Colony. (**b**,**c**) Conidia. (**d**) Conidia with filamentous appendages. (**e**) Adhesive networks. (**f**) Conidiophores. Scale bars: (**a**) = 1 cm, (**b**–**f**) = 20 µm.

#### 3.2.5. *Arthrobotrys yangjiangensis* F. Zhang, S. Boonmee, and X.Y. Yang sp. nov. (Figure 6)

Index Fungorum number: IF901488; Facesoffungi number: FOF14178

Etymology: The species name “*yangjiangensis*” refers to the name of the sample collection site: Yangjiang Town, Yangbi County, Dali City, Yunnan Province, China.

Material examined: China, Yunnan Province, Dali City, Yangbi County, Yangjiang Town, Yangbi River, 25°45′52.11″ N, 99°54′46.43″ E, from freshwater sediment, 14 May 2018, F. Zhang. Holotype CGMCC 3.20968, preserved in the China General Microbiological Culture Collection Center. Ex-type culture DLUCC 124, preserved in the Dali University Culture Collection. Additional specimen examined: China, Yunnan Province, Dali City, Yangbi County, Yangjiang Town, Yangbi River, 25°45′52.11″ N, 99°54′46.43″ E, from freshwater sediment, 14 May 2018, F. Zhang. Living culture YB19, preserved in germplasm resources center of Institute of Eastern-Himalaya Biodiversity Research, Dali University.

Culture characteristics: *Colonies* on PDA white, cottony, reaching 55 mm diameter after 10 days in the incubator at 26 °C. *Hypha* composed of septate, branched, smooth, and hyaline. *Conidiophores* erect, septate, hyaline, unbranched, bearing a single conidium at the apex, 198–537 µm long ($\bar{X}$ = 409.4 µm, n = 50), 3–5.5 µm ($\bar{X}$ = 4.7 µm, n = 50) wide at the base, and 2–3.5 µm ($\bar{X}$ = 2.6 µm, n = 50) wide at the apex. *Conidia* smooth-faced and hyaline, rounded at the apex and truncated at the base, 24.5–47 × 14.5–29.5 µm ($\bar{X}$ = 36.2 × 23.2 μm, n = 50), immature conidia drop-shaped, obovate, with a super cell at the apex and one to two septa (mostly two-septate) at the base; mature conidia broad fusiform, two- to three-septate (mostly three-septate, one septum at the apex and two septa at the base), with a super cell at the middle of the conidia. Conidia germinate from the only small cells at both ends, and the super cells never germinate. Catching nematodes with *adhesive networks*. *Chlamydospores* 8.5–24 × 7–13.5 µm ($\bar{X}$ = 14 × 10 μm, n = 50), smooth-faced and hyaline, cylindrical, globose or ellipsoidal, hyaline, and in chains when present.

Notes: Phylogenetically, *A. yangjiangensis* formed a basal lineage with *A. jinshaensis* (another new species reported in this study), *A. mangrovispora*, *A. thaumasia*, *A. eudermata*, and *A. janus* with 89% MLBS and 0.91 BYPP support (Figure 1). A comparison of *ITS* nucleotides reveled *A. yangjiangensis* was 7.9% (49/622 bp), 12.2% (92/754 bp), 7% (42/604 bp), 9.4% (75/795 bp), and 6.3% (36/573 bp) different from *A. jinshaensis*, *A. mangrovispora*, *A. thaumasia*, *A. eudermata,* and *A. janus*, respectively. Morphologically, *A. yangjiangensis* can be easily distinguished from *A. jinshaensis* and *A. janus* by its conidial shape with drop-shaped, broad fusiform, two- to three-septate conidia [4,39]. The distinguishing characteristics between *A. yangjiangensis*, *A. mangrovispora*, and *A. thaumasia* is that the conidiophores of *A. yangjiangensis* bear only one single conidium at the apex, while the conidiophores of the latter two species produce several short denticles at the apex and bear multiple conidia [4,37,40]. By contrast, *A. yangjiangensis* is more similar to *A. eudermata* in its simple conidiophores and broad fusiform conidia. However, the conidia of *A. yangjiangensis* are noticeably smaller than those of *A. eudermata* (*A. yangjiangensis*, 2.5–4 (36.2) × 14.5–29.5 (23.2) µm versus *A. eudermata*, 37–55 (49) × 17.5–35 (28) µm). Furthermore, *A. eudermata* produces ellipsoid, aseptate microconidia, whereas *A. yangjiangensis* does not [4,41].

**Figure 6 jof-10-00081-f006:**
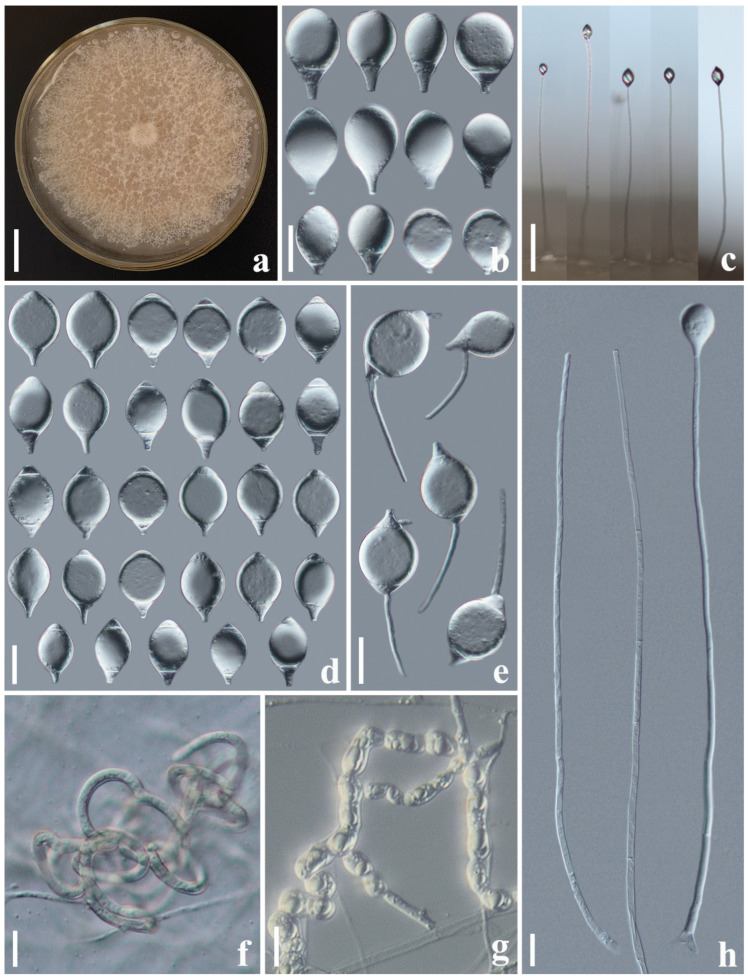
*Arthrobotrys yangjiangensis* (CGMCC 3.20968). (**a**) Colony. (**b**) Immature conidia. (**d**) Mature conidia. (**e**) Germinating conidia (**f**) Adhesive networks. (**g**) Chlamydospores. (**c**,**h**) Conidiophores. Scale bars: (**a**) = 1 cm, (**b**,**d**–**h**) = 20 µm, (**c**) = 100 µm.

## 4. Discussion

Among the 130 species (plus our five new species) of *Orbiliomycetes* nematode-trapping fungi (NTF), 61 species have distribution records in freshwater habitats. Among these species, 17 species were first isolated from freshwater habitats. To date, *A. hyrcanus*, *A. blastospora*, *A. dainchiensis*, *A. eryuanensis*, *A. hengjiangensis* and five new species reported in this study were exclusively found in freshwater environments [12,13,14,15,16,17,34,35,42]. These findings collectively emphasize that freshwater NTF are an important component of NTF diversity. Hence, future studies on NTF diversity should fully consider the significance of freshwater habitats.

The five newly identified species presented were all derived from sediment samples in water depths of less than 2 m. Previous investigations have demonstrated the absence of NTF beyond a water depth of 4 m [13]. However, abundant aquatic nematodes still exist in deeper waters [43,44]. At such depths, there must be other nematode regulators to perform the function of aquatic nematode population regulation instead of NTF. Accordingly, it is speculated that more novel nematophagous microorganisms may be discovered in deeper waters employing efficacious research methodologies. The exploration and examination of these enigmatic nematophagous microorganisms hold the potential to provide valuable insights into the origins and evolutionary processes of carnivorous microorganisms, while also offering promising prospects for the biological control of detrimental nematodes.

It is generally believed that aquatic NTF originate from terrestrial ecosystems. However, unlike terrestrial environments, NTF face numerous challenges in aquatic habitats. For instance, the interaction between NTF and nematodes relies on processes such as host recognition, generation of trapping structures, invasion and digestion of nematode, etc. These processes are more or less dependent on the transmission of extracellular signaling factors [45,46,47]. However, in aquatic environments, these extracellular signaling molecules are likely to be diluted and lose their corresponding functions. Consequently, how aquatic NTF prey on nematodes in water, how they maintain osmotic balance in water, and how they reproduce and spread in water remain unresolved questions in the field. Investigating these questions can not only deepen our understanding of fungal adaptive evolution but also constitutes an important aspect of the study on the origin and evolution of these extraordinary organisms.

## Figures and Tables

**Figure 1 jof-10-00081-f001:**
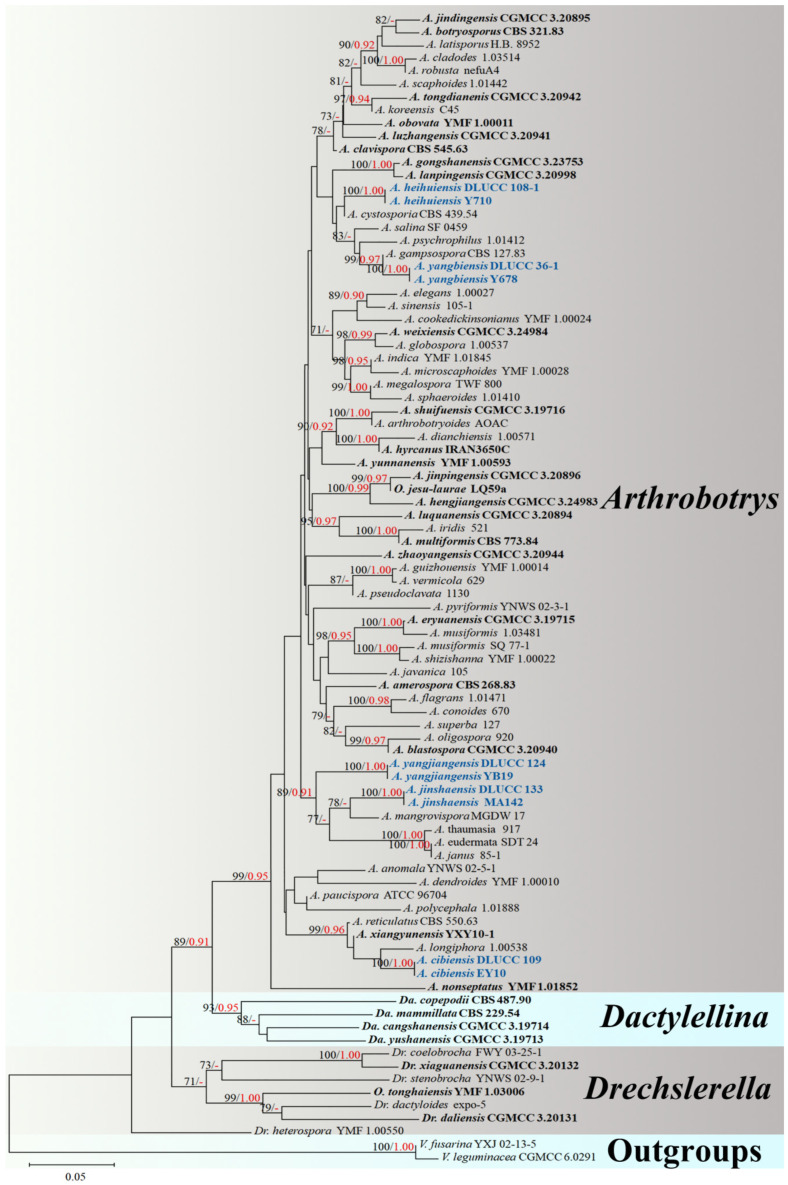
The maximum likelihood tree inferred from a combined *ITS*, *TEF1-α*, and *RPB2* dataset. The black and red numbers in front of the node indicate Bootstrap support values for maximum likelihood equal or greater than 70% and Bayesian posterior probabilities values equal or greater than 0.90, respectively. Our new isolates are in blue and the type strains are in bold.

**Table 1 jof-10-00081-t001:** Samples information involved in this study.

Sample Source	Sampling Location	Sampling Date	Number of Samples
Cibi Lake	26°9′7.14″ N, 99°56’32.72″ E	4 June 2013	25
Heihui River	25°37′4.13″ N, 100°1′52.06″ E	6 April 2018	10
Jinsha River	27°8′50.56″ N, 99°49′39.43″ E	9 July 2014	10
Yangbi River	25°42′37.94″ N, 99°54′52.15″ E	4 April 2018	10
Yangjiang River	25°45′52.11″ N, 99°54′46.43″ E	14 May 2018	10

## Data Availability

The data that support the finding of this study are contained within the article.

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
