# Peer review of "Multilocus Phylogeny and Characterization of Five Undescribed Aquatic Carnivorous Fungi (Orbiliomycetes)"

_jof, 2024, doi:10.3390/jof10010081_

Round 1
Reviewer 1 Report
Comments and Suggestions for Authors
Dear authors
Congratulations for the manuscrpt that presents the interesting and fundamental description of carnivorous fungi.
I have noted some minor editing that I hope will help to improve reader's experience:
In page 3, line 108: a citation is marked in yellow and I do not understand why. Please clarify.
In page 4, Fig. 1, I would like to have the use of bold in some names clarified.
In page 163, I detected the first appearence of "super cell" but the explanation of this word is made in page 7 line 204-205. Please, add the explanation agter the first citation.
In Discussion, I would like to have a discussion on the phylogeny of the 10 species exclusively found in freshwater; their relationship to the 17 species found in freshwater and other substrates. Are they the same? Where are them in Fig 1?
I believe discussion would benefit from a view of the ecology and phylogenetics of those species, among the 61 found in freshwaters. From the manuscript we could not understand if those species are a cluster in the group of carnivorous fungi. In truth, I could not know the list of those 10 species exclusive from freshwater, and also which of them are included in the 17 found first in freshwaters... Are those 10 included in the 17 or not? Are they siblings? What about the 61 species occurring in freshwaters: are the 10 and 17 among those? And the others, are they closely related or not?
Since this is a paper on description of new species I find it important to put these fungi in the wider view of the phylogeny and ecology of the group.
Congratulations for the findings!
Comments on the Quality of English LanguageI point, also, these minor language mistakes:
In page 7 line 212 the 'in' should be deleted
In page 7 line 215 the sentence should be "... heihuiensis are usually inserted'
In page 9 line 246, a comma should be deleted
In page 11 line 290, the correct sentence is ' a cluster of short denticles'
In page 13, line 350, please delete the word sample and correct the word However
Reviewer 2 Report
Comments and Suggestions for Authors
Summary:
- The paper describes five new species of aquatic nematode-trapping fungi isolated from freshwater sediments in Yunnan Province, China.
- The new species belong to the genus Arthrobotrys and are named A. cibiensis,
A. heihuiensis, A. jinshaensis, A. yangbiensis, and A. yangjiangensis.
- Morphological characteristics and phylogenetic analyses based on ITS, TEF1-α and RPB2 sequences are provided to delineate and validate the new species.
- The new fungi display adhesive network traps to catch nematodes. Their conidiophores, conidia, and other microscopic structures are described and imaged in detail.
Main findings:
- Aquatic habitats harbor a rich diversity of nematode-trapping fungi that remains understudied. The discovery of the five new species underscores the need for further investigations into aquatic NTF diversity.
- Phylogenetic analyses placed the new species in distinct clades, confirming their unique taxonomic status compared to known species.
Major comments:
I suggest that the author test whether these 5 species form traps in liquid culture, since they are aquatic fungi.
Comments on the Quality of English Language
The English quality is ok.
